# Uncertainty quantification using Bayesian neural networks in classification: Application to ischemic stroke lesion segmentation

**Yongchan Kwon**
Department of Statistics
Seoul National University
ykwon0407@snu.ac.kr

**Joong-Ho Won**
Department of Statistics
Seoul National University
wonj@stats.snu.ac.kr

**Beom Joon Kim**
Department of Neurology and Cerebrovascular Center
Seoul National University Bundang Hospital
kim.bj.stroke@gmail.com

**Myunghee Cho Paik**
Department of Statistics
Seoul National University
myungheechopaik@snu.ac.kr

## Abstract

Most recent research of neural networks in the field of computer vision has focused on improving accuracy of point predictions by developing various network architectures or learning algorithms. Uncertainty quantification accompanied by point estimation can lead to a more informed decision, and the quality of prediction can be improved. In medical imaging applications, assessment of uncertainty could potentially reduce untoward outcomes due to suboptimal decisions. In this paper, we invoke a Bayesian neural network and propose a natural way to quantify uncertainty in classification problems by decomposing predictive uncertainty into two parts, aleatoric and epistemic uncertainty. The proposed method takes into account discrete nature of the outcome, yielding correct interpretation of each uncertainty. We demonstrate that the proposed uncertainty quantification method provides additional insight to the point prediction using images from the Ischemic Stroke Lesion Segmentation Challenge.

## 1 Introduction

While deep neural networks have shown outstanding performances in many different computer vision tasks and the field of medical imaging analysis [Havaei et al., 2017, Maier et al., 2017, Litjens et al., 2017], less attention has been paid to assessing uncertainty in neural network outputs. Probabilistic interpretations via uncertainty quantification are important because (1) absences of sufficient understanding of model outputs may provide suboptimal results and (2) neural networks are subject to overfitting, so making decisions based on point prediction alone may provide incorrect classifications with spuriously high confidence [Su et al., 2017].

In this paper, we utilize Bayesian neural network models to capture uncertainties in classification problems. Bayesian methods are mathematically sophisticated and allow rich probabilistic interpretations via obtaining predictive distribution. Although exact Bayesian inferences have been considered to be computationally intractable, Gal [2016] recently showed that a typical optimization of dropout neural networks is equivalent to Bayesian learning via variational inference with a specific variational distribution. This provided a practical solution to estimate predictive uncertainty.

In Bayesian modeling, uncertainties are characterized as either aleatoric uncertainty capturing noise inherent in the observations or epistemic uncertainty which accounts for model uncertainty

1st Conference on Medical Imaging with Deep Learning (MIDL 2018), Amsterdam, The Netherlands.

[Der Kiureghian and Ditlevsen, 2009]. Kendall and Gal [2017] decomposed the predictive uncertainty into the two types of uncertainties by explicitly modeling the variability of the last layer of neural network outputs. However, existing studies on the uncertainty quantification for classification have utilized extra parameters for variances without reflecting the relationship between the variance and the mean.

Our main contributions of this paper are:

1. We propose a new method of quantifying uncertainties in classification using Bayesian neural networks. Our method exploits the relationship between the variance and the mean of a multinomial random variable and avoids estimation of extra parameters for the variance.

2. We demonstrate the proposed method using the two Ischemic Stroke Lesion Segmentation Challenge (ISLES) datasets with different effective sample sizes. Our results demonstrate interpretable uncertainty maps, and exhibit dependence between the epistemic uncertainty and the effective sample size.

## 2  Bayesian neural networks

Let $\mathcal{D} = \{(x_i, y_i)\}_{i=1}^N$ be a realization of independently and identically distributed random variables where $x_i \in \mathcal{R}^d$ and $y_i = (y_i^{(1)}, \ldots, y_i^{(K)}) \in \{0, 1\}^K$ are the $i$th input and its corresponding one-hot encoded categorical output, respectively. Here, $N$ denotes the sample size, $d$ is the dimension of input variables, and $K$ is the number of different classes. Assuming the Bayesian neural network model, we place a prior distribution $p(\omega)$ on a parameter vector $\omega \in \Omega$, weights and bias vectors in a neural network, resulting posterior distribution

$$p(\omega \mid \mathcal{D}) = \frac{p(\mathcal{D} \mid \omega)p(\omega)}{p(\mathcal{D})} = \frac{\prod_{i=1}^N p(y_i \mid x_i, \omega)p(\omega)}{p(\mathcal{D})},$$

and predictive distribution

$$p(y^* \mid x^*, \mathcal{D}) = \int_\Omega p(y^* \mid x^*, \omega)p(\omega \mid \mathcal{D})d\omega,$$

for a new input $x^*$ and a new output $y^*$. Denoting by $f^\omega(x) = (f_1^\omega(x), \ldots, f_K^\omega(x))$ the last $K$-dimensional pre-activated linear output of the neural network with a parameter vector $\omega$, then the predictive probability is given by

$$p\{y^{(k)} = 1 \mid x, \omega\} = p\{y^{(k)} = 1 \mid f^\omega(x)\} = \frac{\exp\{f_k^\omega(x)\}}{\sum_{j=1}^K \exp\{f_j^\omega(x)\}}.$$

While this formalization is simple, the learning is often tricky because calculating the posterior $p(\omega \mid \mathcal{D})$ requires an integration with respect to the whole parameter space $\Omega$ for which a closed form often does not exist. MacKay [1992] proposed the a Laplace approximation of the posterior but performance is often limited due to poor approximation.

Neal [1993] introduced the Hamiltonian Monte Carlo, a Markov Chain Monte Carlo (MCMC) sampling approach using Hamiltonian dynamics, to learn Bayesian neural networks. This yields a principled set of posterior samples without direct calculation of the posterior but computationally prohibitive.

An alternative Bayesian method is a variational inference [Graves, 2011, Blundell et al., 2015, Louizos and Welling, 2016, 2017a] which approximates the posterior distribution by a tractable variational distribution $q_\theta(\omega)$ indexed by a variational parameter $\theta$. The optimal variational distribution is the closest distribution to the posterior among the pre-determined family $Q = \{q_\theta(\omega)\}$. The closeness is often measured by the Kullback-Leibler (KL) divergence between $q_\theta(\omega)$ and $p(\omega \mid \mathcal{D})$ defined by

$$KL\{q_\theta(\omega)\|p(\omega \mid \mathcal{D})\} := \int_\Omega q_\theta(\omega) \log \frac{q_\theta(\omega)}{p(\omega \mid \mathcal{D})} d\omega.$$

Minimizing the KL divergence is equivalent to minimizing the negative evidence lower bound given by

$$-\int_\Omega q_\theta(\omega) \log p(y \mid x, \omega)d\omega + KL\{q_\theta(\omega)\|p(\omega)\}. \tag{1}$$

Variational inference converts standard Bayesian learning from integration to optimization problem. This conversion allows online learning in Bayesian inference which is originally suited for batch learning due to MCMC sampling.

Since variational distribution $q_\theta(\omega)$ approximates to the posterior, the quality of approximation depends on the family of distributions $Q$. Restricted family $Q$ would help scalability but hurt approximation. Graves [2011] and Blundell et al. [2015] specified $q_\theta(\omega)$ as product of normal distribution invoking a mean field approximation. The mean field approximation with normal distributions makes the problem scalable but normality assumption doubles the number of parameters due to mean and variance, which makes optimization more challenging. Recently, Gal [2016] proved that a dropout neural network is equivalent to a specific variational approximation in a Bayesian neural network. With this justification, they proposed a method to estimate predictive uncertainty through variational distribution.

Kendall and Gal [2017] further improved the method of Gal [2016] by decomposing the source of uncertainty into aleatoric and epistemic where the former captures irreducible variability due to randomness of outcomes, and the latter, variability arising from estimation. The decomposition was devised through the last layer with extra nodes to learn the variance. Our approach is to construct a variational inference based on the arguments of Gal [2016] and aim to decompose the source of uncertainty without adding extra components to learn the variance as in Kendall and Gal [2017].

## 3 Uncertainty estimation in classification

### 3.1 Aleatoric uncertainty and epistemic uncertainty

Though the proposed method can be applied to any Bayesian inferences, we present the followings in the context of variational inference to make a comparison with the previous study [Kendall and Gal, 2017]. Let $\hat{\theta}$ be the optimized variational parameter iteratively minimizing Monte Carlo approximated version of the equation (1) [Ranganath et al., 2014]. At inference, we derive the variational predictive distribution approximating the predictive distribution

$$q_{\hat{\theta}}(y^* \mid x^*) = \int_\Omega p(y^* \mid x^*, \omega) q_{\hat{\theta}}(\omega) d\omega,$$

then its estimator

$$\hat{q}_{\hat{\theta}}(y^* \mid x^*) = \frac{1}{T} \sum_{t=1}^{T} p(y^* \mid x^*, \hat{\omega}_t)$$

converges in probability, where a set of realized vectors $\{\hat{\omega}_t\}_{t=1}^{T}$ is randomly drawn from variational distribution $q_{\hat{\theta}}(\omega)$ with the pre-defined sampling number $T$. Variance of the variational predictive distribution $q_{\hat{\theta}}(y^* \mid x^*)$ is given by

$$
\begin{aligned}
Var_{q_{\hat{\theta}}(y^*|x^*)}(y^*) &= E_{q_{\hat{\theta}}(y^*|x^*)}\{y^{*\otimes 2}\} - E_{q_{\hat{\theta}}(y^*|x^*)}(y^*)^{\otimes 2}, \\
&= \underbrace{\int_\Omega [\mathrm{diag}\{E_{p(y^*|x^*,\omega)}(y^*)\} - E_{p(y^*|x^*,\omega)}(y^*)^{\otimes 2}] q_{\hat{\theta}}(\omega) d\omega}_{\text{aleatoric}} \\
&\quad + \underbrace{\int_\Omega \left\{ E_{p(y^*|x^*,\omega)}(y^*) - E_{q_{\hat{\theta}}(y^*|x^*)}(y^*) \right\}^{\otimes 2} q_{\hat{\theta}}(\omega) d\omega}_{\text{epistemic}},
\end{aligned}
\tag{2}
$$

where $v^{\otimes 2} = vv^T$, $E_{q_{\hat{\theta}}(y^*|x^*)}\{g(y^*)\} = \int g(y^*) q_{\hat{\theta}}(y^* \mid x^*) dy^*$, $E_{p(y^*|x^*,\omega)}\{g(y^*)\} = \int g(y^*) p(y^* \mid x^*, \omega) dy^*$ for any measurable function $g$, and $\mathrm{diag}(v)$ is a diagonal matrix with elements of the vector $v$. The first equation is obtained by definition of variance, and the second one is from a variant of law of the total variance. The detailed derivation is provided in Appendix A.

Under Bayesian framework, $\omega$ is random, and whenever evaluation of $y^*$ is made for given $x^*$, different features are formed and each feature is weighed differently to determine $p(\omega) = \mathrm{Softmax}\{f^\omega(x^*)\}$. The predictive uncertainty is $E_{q_{\hat{\theta}}(y^*|x^*)}\{y^{*\otimes 2}\} - E_{q_{\hat{\theta}}(y^*|x^*)}(y^*)^{\otimes 2}$. The

aleatoric uncertainty, $E[\text{diag}\{p(\omega)\} - p(\omega)^{\otimes 2}]$ where the expectation is over $q_{\hat{\theta}}(\omega)$, captures inherent randomness of an output $y^*$. The epistemic uncertainty, $E[p(\omega) - E\{p(\omega)\}]^{\otimes 2}$, comes from the variability of $\omega$ given data. This quantity reduces as the sample size increases. That is, the classifier's weighting scheme becomes less variable as the sample size increase.

## 3.2 Proposed method

Kendall and Gal [2017] developed a novel way to directly estimate these two types of uncertainties. They constructed a Bayesian neural network model with the last layer before activation consisting of mean and variance of logits. We denote by $f^{\omega}_{kendall}(x^*) = (\mu, \sigma^2)$ the last $2K$-dimensional pre-activated linear output of the neural network, where $\mu$ and $\sigma^2$ represent the mean and the variance of $K$ nodes. For the realized vectors $\{\hat{\omega}_t\}_{t=1}^T$ and the corresponding outputs $f^{\hat{\omega}_t}_{kendall}(x^*) = (\hat{\mu}_t, \hat{\sigma}_t^2)$, they suggested an estimator of two types of uncertainty as

$$\underbrace{\frac{1}{T}\sum_{t=1}^T \text{diag}(\hat{\sigma}_t^2)}_{\text{aleatoric}} + \underbrace{\frac{1}{T}\sum_{t=1}^T (\hat{\mu}_t - \bar{\mu})^{\otimes 2}}_{\text{epistemic}}, \tag{3}$$

where $\bar{\mu} = \sum_{t=1}^T \hat{\mu}_t / T$.

Quantifying uncertainty by the equation (3) has at least two limitations for use with classification. First, the equation (3) models the variability of the linear predictors, not the predictive probabilities and ignores the fact that the covariance matrix of a multinomial random variable is a function of the mean vector. For example, in binary classification, the predicted values closer to 0 or 1 have a small variance. Deep neural networks for classification are generalized linear models with error structure of multinomial and composite link functions, and therefore the variance of the multinomial outcome is a function of the mean. Another limitation is that the aleatoric uncertainty does not reflect correlations due to a diagonal matrix modeling. To solve these problems, we fix the dimension of the last layer as $K$ and propose the estimator for predictive uncertainty by

$$\underbrace{\frac{1}{T}\sum_{t=1}^T \text{diag}(\hat{p}_t) - \hat{p}_t^{\otimes 2}}_{\text{aleatoric}} + \underbrace{\frac{1}{T}\sum_{t=1}^T (\hat{p}_t - \bar{p})^{\otimes 2}}_{\text{epistemic}}, \tag{4}$$

where $\bar{p} = \sum_{t=1}^T \hat{p}_t / T$ and $\hat{p}_t = p(\hat{\omega}_t) = \text{Softmax}\{f^{\hat{\omega}_t}(x^*)\}$. The proposed method directly computes the variability of the predictive probability and does not involve extra $\sigma^2$ term. In the computational aspects, the proposed method does not need extra sampling steps to obtain prediction. By eliminating extra parameters $\sigma^2$, we can reduce the number of parameters and operations. The two terms in equation (4) respectively converge in probability to equation (2) as $T$ increases.

## 4 Experimental results

In this section, we use the public medical imaging datasets from Ischemic Stroke Lesion Segmentation (ISLES) Challenges to demonstrate the merit of the proposed method.

### 4.1 Datasets

The International Conference on the Medical Image Computing and Computer-Assisted Intervention (MICCAI) is hosting the ISLES challenges annually since 2015. We restrict our focus to the comparison of the two datasets from the 2015 ISLES Challenges. One is sub-acute ischemic stroke lesion segmentation (SISS), and the other is acute stroke perfusion estimation (SPES). The main task for both competitions is to construct an automatic ischemic stroke lesion segmentation model. For both competitions, multiple 3-dimensional magnetic resonance imaging (MRI) sequences and ground truth locating stroke lesion are given. Comparison of the two datasets are briefly summarized in Table 1 and detailed information can be found in Maier et al. [2017].

|  | SISS | SPES |
|---|---|---|
| Number of subjects | 28 | 30 |
| Total number of voxels | 227 M | 22 M |
| Average size of the target image | $230 \times 230 \times 154$ | $96 \times 110 \times 72$ |
| MRI sequences | DWI, FLAIR T1, T2 | TTP, Tmax, CBV, CBF T1c, T2, DWI |
| Proportion of stroke lesion Mean (Standard devitation) | 0.54% (0.73%) | 2.51% (1.11%) |

M, million; DWI, diffusion weighted imaging; FLAIR, fluid attenuation inversion recovery; T1, T1-weighted; T2, T2-weighted; TTP, time-to-peak; Tmax, time-to-max; CBV, cerebral blood volume; CBF, cerebral blood, flow; T1c, T1 contrast enhanced.

Table 1: Comparison of the SISS and SPES datasets.

## 4.2 Evaluations

First, we applied the proposed method and that by Kendall and Gal [2017] to SISS and SPES datasets. The implementation details are provided in Appendix B. Figure 1 presents a slice of original MRI image, estimates of aleatory, epistemic, and the sum, labeled by predictive uncertainties along with the predicted lesion and the ground truth. We present the Flair and DWI images as the original MRI image from SISS and SPES dataset, respectively, since the ground truth segmentation was made from respective imaging modes. These images are known to represent the core area of stroke well [Maier et al., 2017]. Bright color represents large values. The rows labeled by 'P' and 'K' indicate the proposed and the method by Kendall and Gal [2017], respectively.

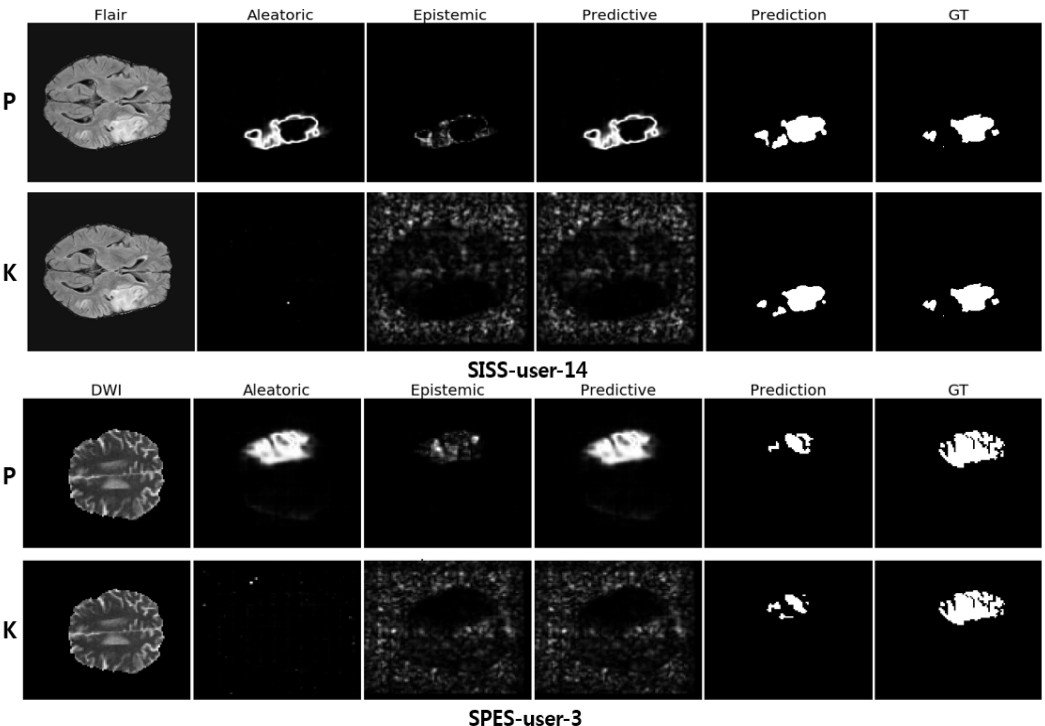

Figure 1: Visual comparisons of uncertainty quantification methods using the user '14' of the SISS dataset (row 1 and 2) and the user '3' of the SPES dataset (row 3 and 4). Row 'P' and 'K' indicate the proposed method and the uncertainty quantification method by Kendall and Gal [2017], respectively.

Although both methods produced similar predicted values, there were substantial differences in the uncertainty maps. In our experiments, the aleatoric uncertainty terms, the extra parameter in Kendall and Gal [2017] converged to zero and uncertainty image maps produced by Kendall and Gal [2017] provided little information. The proposed maps show aleatoric, epistemic and combined uncertainties.

First, the main source of uncertainty came from the aleatoric part. Second, the boundaries were shown to be more uncertain than the interior regions. In the images shown in Figure 1, the uncertainty maps provided extra information in addition to the prediction map. When the prediction map disagrees with the ground truth, the uncertainty maps identified the stroke region missed by prediction map. In SISS-user-14, the proposed prediction map incorrectly identified non-stroke region as stroke, and the uncertainty map reflected a lack of confidence around the incorrectly identified region. The prediction map from the SPES-user-3 partially failed to identify some lesion locations but the uncertainty maps recovered the missed region close to the ground truth. More visual results are provided in Appendix C.

### 4.3 Implication of difference in resolution or voxel numbers

We conducted a comparison study using the proposed equation (4) between SISS and SPES datasets. The main difference between the two datasets is in the number of voxels originating from the disparate resolutions of the images. The number of subjects is comparable for SISS and SPES. In segmentation tasks, unit of analysis is a voxel and the outcomes constitute high-dimensional multivariate binary random variables. Since the voxels are not independent, the number of voxels does not directly translate to the sample size. On the other hand, the voxels are not perfectly correlated, and the effective sample size is between the number of voxels and the number of cases [Fleiss et al., 2003]. In this regard, the effective sample size is bigger in SISS than SPES. Recall that aleatoric uncertainty is irreducible but the epistemic uncertainty decreases with the sample size. Therefore we anticipate the epistemic uncertainty to be smaller for SISS.

To verify this, we examined bivariate density plots for the aleatoric and epistemic uncertainties among the top 1% data points with largest aleatoric uncertainty. Figure 2 shows bivariate density plots of SISS on the left and SPES on the right. Quantified uncertainty tends to be smaller for SISS than SPES. The SISS dataset also exhibits smaller epistemic uncertainty than SPES as anticipated. Similar trends were observed when using the top 5% data.

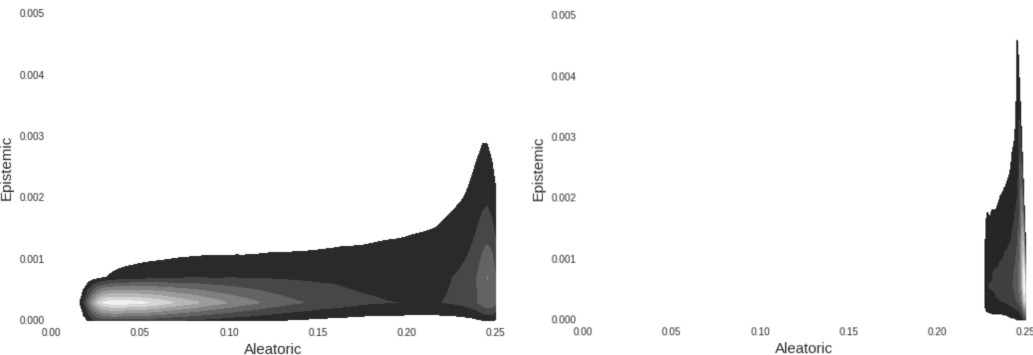

Figure 2: Bivariate density plots for aleatoric and epistemic uncertainties among the top 1% data points in terms of the aleatoric uncertainty. Left, SISS, and right, SPES datasets.

Table 2 compares the mean of the aleatoric and epistemic uncertainties of SPES and SISS. Five-folds cross-validation was used for all computations for a fair comparison between the two datasets. The aleatoric uncertainty of SPES is about five times as large as that of SISS. Considering that the aleatoric uncertainty measures irreducible variability and depends on the predicted values, a large aleatoric uncertainty for SPES may be due to that the marginal proportion is larger for SPES than SISS as shown in Table 1. The epistemic uncertainty of SISS is smaller than that of SPES as anticipated since the epistemic uncertainty decreases with the effective sample size.

|  | SISS | SPES |
|---|---|---|
| Aleatoric | $7.6 \times 10^{-4}$ | $3.6 \times 10^{-3}$ |
| Epistemic | $2.2 \times 10^{-5}$ | $1.4 \times 10^{-4}$ |

Table 2: Comparison of the averages of aleatoric and epistemic uncertainties of the SISS and SPES datasets.

Since the variability of a binary random variable depends on the mean, the difference in the aleatoric uncertainty can confound the comparison of the epistemic uncertainty. To circumvent this problem, we compared the estimated conditional expectations of the epistemic uncertainty given ranges of the aleatoric uncertainty. The results are given in Table 3. Although the differences are small, there is a coherent trend that the epistemic uncertainty is smaller for the SISS dataset than for the SPES dataset by 5-10%.

|  | SISS | SPES |
|---|---|---|
| $E(\text{Epistemic} \mid 0.05 < \text{Aleatoric} < 0.1)$ | $4.4 \times 10^{-4}$ | $4.6 \times 10^{-4}$ |
| $E(\text{Epistemic} \mid 0.1 < \text{Aleatoric} < 0.15)$ | $8.9 \times 10^{-4}$ | $9.4 \times 10^{-4}$ |
| $E(\text{Epistemic} \mid 0.15 < \text{Aleatoric} < 0.2)$ | $13.8 \times 10^{-4}$ | $14.5 \times 10^{-4}$ |
| $E(\text{Epistemic} \mid 0.2 < \text{Aleatoric} < 0.25)$ | $17.6 \times 10^{-4}$ | $19.4 \times 10^{-4}$ |

Table 3: Comparison of the conditional expectation of epistemic uncertainty given an aleatoric intervals for the SISS and SPES datasets.

## 5   Conclusion and discussion

In this paper, we presented a natural way to quantify uncertainty of classification in Bayesian neural networks by decomposing the uncertainty into two types. The proposed method has advantages in that the inherent variability is expressed in terms of the underlying distribution of the outcome and has the practical advantage of numerical stability. Application of our method to ISLES datasets provided additional insights into the corresponding medical imaging analysis with point estimation alone.

We acknowledge that variational variability may not be a good approximation of the posterior variance. However, the epistemic uncertainty can be useful in comparing the variability due to the effective sample size. Our comparison study between the two datasets with differential information supports the utility of the epistemic uncertainty.

The proposed method of uncertainty quantification is applied in the variational inference setting in this paper. However, it can be applied to general Bayesian setting and to broader variational inference problems. For example, the equation (4) can be exploited when the class of family $Q$ is enlarged as in [Miller et al., 2016, Louizos and Welling, 2017b], or when different divergence metrics are used ([Li and Gal, 2017, Dieng et al., 2017]). Extension of the proposed method to different settings can be future research topics.

## 6   Acknolwedgement

This paper extends the paper, 'Uncertainty quantification for ischemic stroke lesion segmentation using Bayesian neural networks', published at Neural Information Processing Systems 2017 Workshop on 'Medical Imaging meets NIPS'. The authors were supported by the National Research Foundation of Korea under grant NRF-2017R1A2B4008956.

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

# Appendix: Uncertainty quantification using Bayesian neural networks in classification: Application to ischemic stroke lesion segmentation

**Yongchan Kwon**
Department of Statistics
Seoul National University
ykwon0407@snu.ac.kr

**Joong-Ho Won**
Department of Statistics
Seoul National University
wonj@stats.snu.ac.kr

**Beom Joon Kim**
Department of Neurology and Cerebrovascular Center
Seoul National University Bundang Hospital
kim.bj.stroke@gmail.com

**Myunghee Cho Paik**
Department of Statistics
Seoul National University
myungheechopaik@snu.ac.kr

In this appendix material, we present (A) a derivation of the equation (2), (B) implementation details, and (C) additional visualization results.

## A  Derivation of the equation (2)

Recall that Equation (2) is given by

$$
\begin{aligned}
Var_{q_{\hat{\theta}}(y^*|x^*)}(y^*) &= E_{q_{\hat{\theta}}(y^*|x^*)}\{y^{*\otimes 2}\} - E_{q_{\hat{\theta}}(y^*|x^*)}(y^*)^{\otimes 2}, \\
&= \underbrace{\int_\Omega [\mathrm{diag}\{E_{p(y^*|x^*,\omega)}(y^*)\} - E_{p(y^*|x^*,\omega)}(y^*)^{\otimes 2}]q_{\hat{\theta}}(\omega)d\omega}_{\text{aleatoric}}, \\
&+ \underbrace{\int_\Omega \left\{ E_{p(y^*|x^*,\omega)}(y^*) - E_{q_{\hat{\theta}}(y^*|x^*)}(y^*) \right\}^{\otimes 2} q_{\hat{\theta}}(\omega)d\omega}_{\text{epistemic}},
\end{aligned}
$$

where $v^{\otimes 2} = vv^T$, $E_{q_{\hat{\theta}}(y^*|x^*)}\{g(y^*)\} = \int g(y^*)q_{\hat{\theta}}(y^* \mid x^*)dy^*$, $E_{p(y^*|x^*,\omega)}\{g(y^*)\} = \int g(y^*)p(y^* \mid x^*,\omega)dy^*$ for any measurable function $g$, and $\mathrm{diag}(v)$ is a diagonal matrix with elements of the vector $v$.

*Proof.* The first equation follows from the definition of variance and it is enough to show the second equation. From the definition of $E_{q_{\hat{\theta}}(y^*|x^*)}\{g(y^*)\}$ and by Fubini's Theorem, we have

$$
\begin{aligned}
E_{q_{\hat{\theta}}(y^*|x^*)}(y^*) &:= \int y^* q_{\hat{\theta}}(y^* \mid x^*)dy^*, \\
&= \int y^* \int_\Omega p(y^* \mid x^*,\omega)q_{\hat{\theta}}(\omega)d\omega dy^*, \\
&= \int_\Omega \int y^* p(y^* \mid x^*,\omega)dy^* q_{\hat{\theta}}(\omega)d\omega, \\
&= \int_\Omega E_{p(y^*|x^*,\omega)}(y^*)q_{\hat{\theta}}(\omega)d\omega, \quad\quad\quad\quad\quad (A.1)
\end{aligned}
$$

1st Conference on Medical Imaging with Deep Learning (MIDL 2018), Amsterdam, The Netherlands.

and similarly,

$$E_{q_{\hat{\theta}}(y^*|x^*)}(y^{*\otimes 2}) := \int y^{*\otimes 2} q_{\hat{\theta}}(y^* \mid x^*) dy^*,$$

$$= \int y^{*\otimes 2} \int_\Omega p(y^* \mid x^*, \omega) q_{\hat{\theta}}(\omega) d\omega dy^*,$$

$$= \int_\Omega \int y^{*\otimes 2} p(y^* \mid x^*, \omega) dy^* q_{\hat{\theta}}(\omega) d\omega,$$

$$= \int_\Omega E_{p(y^*|x^*,\omega)}(y^{*\otimes 2}) q_{\hat{\theta}}(\omega) d\omega,$$

$$= \int_\Omega \{ Var_{p(y^*|x^*,\omega)}(y^*) + E_{p(y^*|x^*,\omega)}(y^*)^{\otimes 2} \} q_{\hat{\theta}}(\omega) d\omega,$$

where $Var_{p(y^*|x^*,\omega)}(y^*) := E_{p(y^*|x^*,\omega)}(y^{*\otimes 2}) - E_{p(y^*|x^*,\omega)}(y^*)^{\otimes 2} = E_{p(y^*|x^*,\omega)}\{\mathrm{diag}(y^*)\} - E_{p(y^*|x^*,\omega)}(y^*)^{\otimes 2}$ because $y^*$ is one-hot encoded.

Thus,

$$E_{q_{\hat{\theta}}(y^*|x^*)}\{y^{*\otimes 2}\} - E_{q_{\hat{\theta}}(y^*|x^*)}(y^*)^{\otimes 2},$$

$$= \int_\Omega \{ Var_{p(y^*|x^*,\omega)}(y^*) + E_{p(y^*|x^*,\omega)}(y^*)^{\otimes 2} \} q_{\hat{\theta}}(\omega) d\omega - E_{q_{\hat{\theta}}(y^*|x^*)}(y^*)^{\otimes 2},$$

$$= \int_\Omega Var_{p(y^*|x^*,\omega)}(y^*) q_{\hat{\theta}}(\omega) d\omega + \int_\Omega \{ E_{p(y^*|x^*,\omega)}(y^*)^{\otimes 2} - E_{q_{\hat{\theta}}(y^*|x^*)}(y^*)^{\otimes 2} \} q_{\hat{\theta}}(\omega) d\omega,$$

$$= \int_\Omega [ E_{p(y^*|x^*,\omega)}\{\mathrm{diag}(y^*)\} - E_{p(y^*|x^*,\omega)}(y^*)^{\otimes 2} ] q_{\hat{\theta}}(\omega) d\omega,$$

$$+ \int_\Omega \{ E_{p(y^*|x^*,\omega)}(y^*)^{\otimes 2} - E_{q_{\hat{\theta}}(y^*|x^*)}(y^*)^{\otimes 2} \} q_{\hat{\theta}}(\omega) d\omega,$$

$$= \int_\Omega [ \mathrm{diag}\{ E_{p(y^*|x^*,\omega)}(y^*) \} - E_{p(y^*|x^*,\omega)}(y^*)^{\otimes 2} ] q_{\hat{\theta}}(\omega) d\omega,$$

$$+ \int_\Omega \{ E_{p(y^*|x^*,\omega)}(y^*)^{\otimes 2} - E_{q_{\hat{\theta}}(y^*|x^*)}(y^*)^{\otimes 2} \} q_{\hat{\theta}}(\omega) d\omega,$$

$$= \int_\Omega [ \mathrm{diag}\{ E_{p(y^*|x^*,\omega)}(y^*) \} - E_{p(y^*|x^*,\omega)}(y^*)^{\otimes 2} ] q_{\hat{\theta}}(\omega) d\omega,$$

$$+ \int_\Omega \left\{ E_{p(y^*|x^*,\omega)}(y^*) - E_{q_{\hat{\theta}}(y^*|x^*)}(y^*) \right\}^{\otimes 2} q_{\hat{\theta}}(\omega) d\omega,$$

The second last equation holds from commutativity of a diagonal operation and an expectation. The last the equation is from the equation (A.1).

## B  Implementation details

Our learning process and implementations details mostly followed the method as proposed in section 2.1 of Choi et al. [2016]. We trained neural networks through patch learning and omitted the fine-tuning step which was turned out to be not effective in terms of Dice coefficients. For Bayesian learning, the variational inference with dropout variational distribution [Gal, 2016] was used and the number of realized sets $T$ used in the Monte Carlo integration were five.

Figure B.1 shows our network architecture, the 3D multiscale residual U-Net [Choi et al., 2016] without residual blocks. The total number of parameters were around 293k. For the method by Kendall and Gal [2017], we removed the softmax activation layer and changed the number of channels in the last convolutional layer from 1 to 2. Then, we randomly drew a sample from normal distribution using the network outputs, and used it to compute numerically-stable stochastic loss, which is a numerical stable version of a negative log likelihood function, given by Kendall and Gal [2017].

We used the following hyper-parameters. All the MRI sequences were resized to the same dimension: (width $\times$ height $\times$ depth) = ($100 \times 100 \times 72$). For pre-processing step, we extracted 600 patches

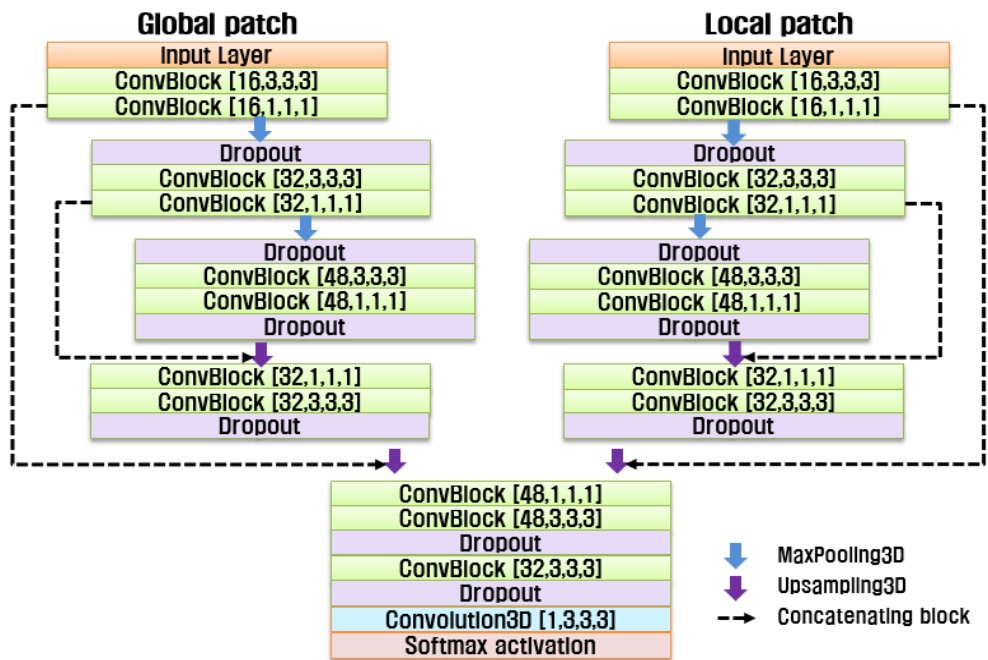

Figure B.1: Visual description of the main network architecture. The 'ConvBlock' is made up of three dimensional convolutional layer, ELU activation [Clevert et al., 2015], and batch normalization layer [Ioffe and Szegedy, 2015]. The numbers in parenthesis denote the number of channels and kernel size in three dimension.

with the size of (width $\times$ height $\times$ depth) = ($16 \times 16 \times 16$). All the weights were initialized as [He et al., 2015]. The Adam optimizer with the learning rate of $0.0002$ [Kingma and Ba, 2014] and an early stopping rule with 10-epoch patience were applied and batch size was 16. The negative log likelihood function was used as a loss function. The Implementations utilized Keras and Theano. The main Python scripts are public at `http://github.com/ykwon0407/UQ_BNN`.

## C  Additional visualization results

In Figure C.1, we presented additional visualization results comparing proposed method with the method by [Kendall and Gal, 2017]. Row 'K' and 'P' indicate the uncertainty quantification method by Kendall and Gal [2017] and the proposed method, respectively. In multiple examples, the proposed method demonstrates informative uncertainty maps.

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
