# OpenReview forum: "Uncertainty quantification using Bayesian neural networks in classification: Application to ischemic stroke lesion segmentation"
_MIDL.amsterdam/2018/Conference — MIDL 2018 Oral_

### Review · AnonReviewer2 · 2018-05-09
**Measuring the uncertainty of a segmentation that is done by a neural network. The problem of quantifying uncertainty of a lesion segmentation that is done by a neural network is an important goal.**

**Rating:** 4
**Confidence:** 2

**Review:**

-	The paper addresses the problem of measuring the uncertainty of a segmentation that is done by a neural network.  It quantifies uncertainty in classification problems by decomposing predictive uncertainty into two parts, aleatoric and epistemic uncertainty.  The proposed method is based on Bayesian neural network and is a variant of a  method proposed by  Kendal and Gal in NIPS 2017.  They use a very similar approach.  The only difference is the parametric model that is used to describe the uncertainty in classification problems. In the original paper they used a Gauassian distribution for the values before the softmax operation. In this paper they use a parametric modeling that is explicitly aware of the discrete nature of the classification operation.


-	The paper is clearly written, and it applies an uptodate deep learning method to medical imaging.  The problem of quantifying uncertainty of a lesion segmentation that is done by a neural network is an important goal. The paper shows several examples (in figure 1) to demonstrate the performance of the proposed method compared to the original methods of Kendall and Gal. What is missing is a quantitative comparison between the two methods that is done in a systematic way on a large dataset.  If the authors could show that their method works better than the baseline the usefulness of it was clearer.



**Special Issue:**

No

---

> ### Comment · ~Yongchan_Kwon1 · 2018-05-22
> **We are thank you for your review.**
>
> For the quantitative comparison, we definitely agree with the reviewer's opinion. We have been thinking of applying the proposed method to multiple datasets including medical imaging sets in future submission to special issue Medical Image Analysis. Thank you for the suggestion.
>
> One thing we want to point out is, though extensive data analysis were not conducted, that the proposed method is still more natural in that it directly measures uncertainties in prediction values.

---

### Review · AnonReviewer1 · 2018-05-09
**Original approach towards uncertainty in classification using DNNs**

**Rating:** 4
**Confidence:** 2

**Review:**

The paper builds up on the work of Kendall ang Gal [2017]. Where it has been proposed to decompose the predictive network uncertainty into model dependent noise (epistemic) and the noise inherent to the data (aleatoric). Authors of the reviewed paper, propose an alternative noise decomposition to Kendall and Gal [2017] for classification, by noting that the variance of the multinomial rv is a function of its mean, with a subsequently provided derivation. Kendall and Gal [2017] model the variability only of a linear predictor, but not the noise of the predictive probabilities as compared to the authors of the reviewed paper, through an additional output layer to predict the data dependent heteroscedastic noise. Derivation of the authors does not require additional output layer, which allows to freely use already existing architectures that allow sampling the predicted probabilities.

Overall, great contribution to the investigation of uncertainty quantification in deep learning and its importance in medical imaging. However, a detailed evaluation of uncertainty quality, as well as the benefits of having predictive uncertainty, over multiple datasets of medical imaging domain would be relevant to fully demonstrate the advantage of the proposed method.

When using a GP for example, we might incorporate input dependent noise using additional GP, making the model heteroscedastic. Which is what is done in the work of Gal and Kendall, where noise is input dependent. If I understand correct the aleatoric noise in your case is not heteroscedastic?


**Special Issue:**

Yes

---

> ### Comment · ~Yongchan_Kwon1 · 2018-05-22
> **Thank you for your thorough review!**
>
> Here is our reply:
>
> 1.
> For the detailed evaluation, we agree with the reviewer's opinion. We have been thinking of applying the proposed method to multiple datasets including medical imaging sets in future submission to special issue Medical Image Analysis. Thank you for the suggestion.
>
> 2.
> The proposed method indeed models heteroscedastic uncertainties since aleatoric uncertainty is a function of a mean, which depends on input data.

---

### Review · AnonReviewer3 · 2018-05-09
**In this paper the authors build on the work of Kendall and Gal for quantifying the aleatoric and epistemic uncertainty for each pixel, in segmentation problems. They improve upon that work by avoiding to learn extra components to model the variance by exploiting that the variance of a multinomial is a function of its mean. That also avoids impossible outcomes like having mean near 0 or 1 in the binomial case and high variance.**

**Rating:** 3
**Confidence:** 2

**Review:**

The paper is well written, the literature related to the paper is thoroughly described, considering the limited amount of space available in the paper. The experiments give some empirical evidences favoring the proposed model, although it is difficult to compare.
It would have been nice to see a couple of plots for aleatoric, and epistemic uncertainty vs frequency of misclassified pixels, to see if any of the estimated uncertainties can predict the probability of misclassified pixels.
It could also be interesting to see how the model performs for different values of T, and how this impacts the inference time.

In any case, given how important uncertainty quantification (of both kinds) is in medical applications, MIDL is a nice venue for this paper, so I recommend this for publication.

**Special Issue:**

Definitely

---

> ### Comment · ~Yongchan_Kwon1 · 2018-05-22
> **Thank you for your kind and encouraging review.**
>
> Regarding different values of $T$ (the number of realized sets), we also noticed that choosing $T$ is a very important issue for two following reasons.
>
> First, in terms of performances, we found prediction results do not significantly change in our experiments. Previous studies related to this can be found in Figure 3 of [1]. Second, in terms of inference time, it linearly increases as $T$ increases. To the best of our knowledge, Bayesian methods inevitably pay this sampling costs and one practical solution is to use parallel computing.
>
> [1] Gal, Yarin, and Zoubin Ghahramani. "Bayesian convolutional neural networks with Bernoulli approximate variational inference." arXiv preprint arXiv:1506.02158 (2015).

---

### Comment · ~Bram_van_Ginneken1 · 2018-05-18
**Selection for longlist for special issue Medical Image Analysis**

Dear authors,

Congratulations on your acceptance to MIDL! We have selected your paper on the longlist for the Medical Image Analysis Special Issue. Please read this page:
https://midl.amsterdam/special-issue-in-medical-image-analysis/
Please answer the three questions that are listed on that page about your interest in submitting to the special issue, potential overlap with other publications, and related publications.

You can post your answer here directly below on openreview.net, or mail me directly at bram.vanginneken@radboudumc.nl.

Best regards, Bram

---

> ### Comment · ~Yongchan_Kwon1 · 2018-05-22
> **Response to the selection for longlist for special issue Medical Image Analysis**
>
> Dear Prof. Bram van Ginneken,
>
> First of all, thanks a lot for selecting our paper 'Uncertainty quantification using Bayesian neural networks in classification: Application to ischemic stroke lesion segmentation'. Here are the answers:
>
> 1. Yes, we will significantly extend our paper and will submit the full manuscript.
>
> 2. We confirm that our paper, or any paper with overlap with the contents of the MIDL paper, is and will not be under review or under consideration elsewhere.
>
> 3. As we stated in the acknowledgment part of the MIDL paper, the two-page extended abstract, which is an early stage version, was published at Neural Information Processing Systems 2017 Workshop on 'Medical Imaging meets NIPS'. Except for the extended abstract, we do not have any related publications of the author group.
>
> Thanks again.
>
> Best regards,
> Yongchan Kwon

---

### Decision · Program_Chairs · 2018-05-15
**Paper11 Acceptance Decision**

Oral